# Genetically Engineered Mice Unveil In Vivo Roles of the Mediator Complex

**DOI:** 10.3390/ijms24119330

**Published:** 2023-05-26

**Authors:** Leonid A. Ilchuk, Marina V. Kubekina, Yulia D. Okulova, Yulia Yu. Silaeva, Victor V. Tatarskiy, Maxim A. Filatov, Alexandra V. Bruter

**Affiliations:** 1Center for Precision Genome Editing and Genetic Technologies for Biomedicine, Institute of Gene Biology, Russian Academy of Sciences, 119334 Moscow, Russia; lechuk12@gmail.com (L.A.I.); kubekina@genebiology.ru (M.V.K.); ul.okulova@gmail.com (Y.D.O.); aleabruter@gmail.com (A.V.B.); 2Institute of Gene Biology, Russian Academy of Sciences, 34/5 Vavilov Street, 119334 Moscow, Russia; yulya.silaeva@gmail.com (Y.Y.S.); tatarskii@gmail.com (V.V.T.); 3Federal State Budgetary Institution “N.N. Blokhin National Medical Research Center of Oncology”, Ministry of Health of the Russian Federation, Kashirskoe Sh. 24, 115478 Moscow, Russia

**Keywords:** Mediator, knockout, transcription regulation, cyclin-dependent kinase, development, metabolism, immunity

## Abstract

The Mediator complex is a multi-subunit protein complex which plays a significant role in the regulation of eukaryotic gene transcription. It provides a platform for the interaction of transcriptional factors and RNA polymerase II, thus coupling external and internal stimuli with transcriptional programs. Molecular mechanisms underlying Mediator functioning are intensively studied, although most often using simple models such as tumor cell lines and yeast. Transgenic mouse models are required to study the role of Mediator components in physiological processes, disease, and development. As constitutive knockouts of most of the Mediator protein coding genes are embryonically lethal, conditional knockouts and corresponding activator strains are needed for these studies. Recently, they have become more easily available with the development of modern genetic engineering techniques. Here, we review existing mouse models for studying the Mediator, and data obtained in corresponding experiments.

## 1. Introduction

The Mediator is a eukaryotic multiprotein complex which functions as a transcriptional coactivator connecting RNA polymerase II and transcriptional factors. The structure of the Mediator complex is evolutionarily conserved, and the mammalian Mediator consists of 26 proteins subdivided into the head (MED6, MED8, MED11, MED17, MED18, MED20, MED22), the middle (MED1, MED4, MED7, MED9, MED10, MED19, MED21, MED26, MED31) and the tail (MED14, MED15, MED16, MED23, MED24, MED25, MED27, MED28, MED29, MED30) parts, and an additional kinase module (Figure 1). The CDK8 kinase module (CKM) is composed of cyclin-dependent kinase CDK8, Cyclin C, MED12, and MED13. Three proteins of the CKM have paralogs—CDK19 is the paralog of CDK8, while MED12L and MED13L are paralogs of MED12 and MED13, respectively.

The current model proposes that the Mediator complex binds to enhancers, while the CKM sterically prevents interaction between the Mediator and RNA Pol II [1]. The kinase module is suspected to negatively regulate the Mediator; however, the precise mechanism is debated [2]. The phosphorylation of core Mediator components and CKM itself by CDK8 leads to dissociation of the CKM module and the binding of the core Mediator, together with MED26, to the PIC and RNA Pol II [3]. The Mediator together with TFIID organizes the preinitiation complex (PIC), allowing the correct action of CDK7, the correct orientation of RNA Pol II, and then the recruitment of other components of PIC (reviewed in [3]). Another hypothesis, based on the yeast experiments, is that MED3 phosphorylation by CDK8 primes the ubiquitination of MED3 and the turnover of the Mediator [4]. Additionally, the CKM is involved in regulating the pausing of transcription, through phosphorylation and interaction with elongation factors—AFF4, NELF and p-TEFB [5]. Additionally, CDK8 and CDK19 phosphorylate a number of transcription factors at enhancers, such as STAT1/3 [6], SMADs [6], and others, modifying their activity. The fact that functions of the Mediator and its CKM are determined by phosphorylation and mechanical interactions lead to the hypothesis that the Mediator can have kinase-dependent and independent functions, and those inhibitors of CDK8/19 would not always phenocopy the knockouts of Mediator components. Although conclusive evidence has not yet been collected, there are data indicating that several members of the mediator complex—MED28 [7] and MED31 [8]—and especially the CKM [9,10,11,12] can perform some functions independent of the Mediator.

Though the Mediator has attracted significant attention because of its role in cancer [13,14], clinical data suggest that mutations in the genes-coding subunits of the Mediator may cause severe developmental disorders (e.g., neurodevelopmental and behavioral disorders and cardiovascular disorders) [15,16,17,18].

The vast majority of the experiments inquiring into Mediator-related molecular mechanisms are performed using yeast, cell lines, and ex vivo models (e.g., [2,19,20], also reviewed in [3]), which have limited use for understanding its role in the development and functioning of the whole tissues and organs.

Genetically engineered mice, both knockouts and mimicking patient mutations, are more suited for this purpose. It appeared, however, impossible to manufacture constitutive knockouts of Mediator subunits, except for less-important paralogs CDK19 and MED12L, due to the embryonic lethality of homozygous knockouts [7,21,22,23,24]. Though this demonstrates the importance of the Mediator in embryonic development, it indicates the necessity of producing conditional knockouts of the Mediator subunits to study their roles in adult organisms.

As far as we know, conditional or viable constitutive knockouts have been manufactured for several Mediator subunits: MED1, MED19, MED20, MED23, MED30, CDK8, Cyclin C, MED12, MED13, and MED13L (Figure 1). Experiments involving these strains revealed the role of the Mediator in several tissues and corresponding biological processes in adult mice. In addition to embryonic development, the most impacted processes are immune response, adipogenesis, metabolism, cardiac functioning, intestine epithelium differentiation, and several others.

Here, we review published data and the data obtained by IMPC (International Mouse Phenotype Consortium) related to the genome-edited mice with modifications in the genes coding the Mediator subunits, and describe their phenotype and the molecular mechanisms underlying observed phenotypes.

## 2. Embryogenesis and Development

Almost all homozygous KO or knockdown models of the large Mediator complex proteins established to this date are embryonically lethal (Figure 2). The only exceptions are KOs of CDK19 and MED12L, apparently less important paralogs of CKM proteins CDK8 and MED12, respectively. *Cdk19*^−/−^ develop normally, are fertile and have unaffected life span [25]. *Med12l KO* have not yet been published, but were developed by the International Mouse Phenotype Consortium and are at least viable [26]. This indicates the high importance of the Mediator for embryonic development, which is probably explained by the frequent transcriptional program switching required for normal embryo development, tissue differentiation, and organogenesis. Despite the agreement about *Med KO*’s embryonic lethality, information about the stages in which the embryonic loss occurs is somewhat controversial. Thus, information given in the mousephenotype.org (accessed on 2 February 2023) database does not always correspond to observations described in peer-reviewed articles, which can contradict each other as well. Partially, this can be explained by different mouse strains and embryo cultivation protocols. In the current section, we review effects of the Mediator proteins on mouse embryo development and discuss observed contradictions. Models discussed in this chapter are briefly summarized in Table 1.

### 2.1. Preimplantation

Preimplantation development includes different stages and processes, such as zygotic genome activation (ZGA), cell division, the formation of connections between cells, etc. The impairment of Mediator complex functions can lead to embryonic lethality even at the preimplantation stage.

In terms of embryonic development time courses, the earliest described embryonic death of in vitro cultured embryos was caused by *Med13* inactivation by Morpholino oligos injection in zygotes, leading to embryonic death at the 2- or 4-cell stage [31]. RNA sequencing revealed that MED13 is required for chromatin remodeling by the esBAF complex, which is necessary for ZGA. Immediately after fertilization, the zygotic genome was transcriptionally inactive. In mice, the activation of the zygotic genome transcription begins in the middle of the one-cell stage (so-called minor ZGA), and a burst of gene activation occurs after the second round of DNA replication (major ZGA) [36]. Thus, the role of MED13 is very important in ZGA because the amounts of accumulated maternal mRNAs are incapable of sustaining development after 2–4 cell stage. Nevertheless, genetic knockouts of MED13 can proceed to further stages of embryonic development, and we will discuss this below.

The next crucial step in preimplantation development is the formation of connections between cells, and therefore morula compaction. It has been reported that *Cdk8* constitutive KO embryos in vitro are unable to perform implantation, and die at embryonic day (E) 3.5 [34]. *Cdk8 KO* embryos contained fragmented blastomeres at E2.5, and their compaction was impaired.

During normal embryonic development, the morula stage is followed by the blastocyst stage. At the blastocyst stage, a very important event occurs—hatching. During the hatching, the blastocyst leaves the protective shell (*zona pellucida*), which surrounds the embryo during preimplantation development allowing the embryo to implant into the uterine wall. At the blastocyst stage, the embryos’ cells begin to differentiate into two compartments: trophectoderm (TE) and inner cell mass (ICM). Trophectoderm develops into a large part of the placenta, and ICM gives rise to the definitive structures of the fetus. During gastrulation, the ICM is divided into two layers of cells: epiblast and hypoblast.

Homozygous *Med20* KO murine embryos die at the blastocyst stage (E3.5) [23] due to the impaired hatching. At the molecular level, the main difference between wildtype (WT) and homozygous KO blastocysts was the abnormal localization of *Nanog* expression, which had to be limited to differentiating epiblast cells, but was ubiquitous in KO embryos. Heterozygous organisms successfully proceeded further in embryo development.

### 2.2. Gastrulation

During gastrulation, the embryo becomes asymmetric, then the primitive streak forms and cells from the epiblast at the primitive streak undergo epithelial-to-mesenchymal transition and ingress at the primitive streak to form the germ layers.

*Med28* KO caused peri-implantation embryo lethality. When heterozygous mice were crossed, a normal (¼) ratio of homozygous embryos was observed in vitro at the blastocyst stage (E3.5). Embryos presumed to be homozygous KOs were found in the uterus at E6.5, with disorganized extraembryonic tissue, and no discernible epiblast. None of the *Med28^−/−^* blastocyst embryos showed well-developed ICM outgrowths when studied in a 2D in vitro culture. Moreover, both intact and cultured KO blastocysts showed a decrease in ICM pluripotency markers *Oct4*, *Sox2*, and *Nanog* [7] compared to the WT. Therefore, MED28 is required to maintain the ICM pluripotency during peri-implantation development. 

*Med12* KO embryos die at early gastrulation (E7.5) and fail to establish an anterior-posterior axis [22]. No embryos have been found at the head-fold stage, and at E7.5 many embryos were arrested at pre-streak stages. Presumably, these embryos cannot induce mesoderm formation, and Wnt/β-catenin signaling in these embryos is abrogated. Other primitive streak markers were also dysregulated. Specifically, *Tbx6* expression was absent. In addition, *Mixl1* expression was severely reduced in most investigated embryos. The absence of β-catenin in *Med12* knockouts leads to the failure of primitive streak formation and the absence of distal visceral endoderm (DVE) migration to form the anterior visceral endoderm (AVE), thus failing to establish an anterior–posterior axis [37]. 

### 2.3. Neurulation and Organogenesis

Gastrulation is followed by neurulation when neural tube formation and closure occurs, and then the longest stage of embryonic development, organogenesis, takes place.

The normal expression of the Mediator complex components is essential for successful embryo development at the organogenesis stage. A lack of or decrease in Mediator complex genes’ expression leads to embryonic lethality at this stage, due to different reasons.

Embryos with disrupted kinase module components, such as MED12, MED13, Cyclin C, and CDK8 fail at organogenesis and die at E9.5–E12.5.

In genetically edited mice with reduced amounts of MED12, development proceeded further than E7.5 but obvious phenotypic anomalies were found at E9.5. No living embryos were found at E10.5. The embryos had defects in neural tube development, a reduction of the first branchial arch, and a complete absence of the second and third branchial arches, cardiac dysfunction, abnormal somitogenesis, and the shortening of the posterior axis. A molecular mechanism of this phenotype is explained by the dysfunction of the Wnt/β-catenin signaling pathway and Wnt/planar cell polarity (PCP) [22].

Morphological anomalies, similar to ones caused by reduced amounts of MED12 and the disruption of WNT/β-catenin, were observed in *snouty* embryos. *Snouty* is a recessive phenotype characterized by craniofacial abnormalities, discovered during ENU (*N*-ethyl-*N*-nitrosourea) mutagenesis screening, which appeared to be a point mutation in the *Med23* gene. The *Med23^sn/sn^* genotype also caused embryonic lethality before E10.5. This phenotype was caused by WNT/β-catenin signaling upregulation in the cranial placodes. It was almost fully phenocopied by the *Med23* constitutive KO. Mutant *Med23^bgeo/bgeo^* embryos (which carried a genetrap between exons 12 and 13 in *Med23* gene) died at E10.5 and failed to undergo developmental turning, and in general showed developmental delay [32]. At the same time, specific KO in the neural crest cells had a milder phenotype and resulted in embryos with craniofacial anomalies surviving until postnatal day 0 (P0) [32]. Vascular defects were also observed in *Med23^sn/sn^* and *Med23* KOs [21,32]. However, embryos with tissue specific *Med23* KO in vascular endothelial cells survived until E16.5, indicating that this was not the main cause of death in constitutive KO [32].

As has been mentioned above, constitutive *Cdk8* KO fails to proceed with preimplantation development [34]. However, recently the conditional knockout of *Cdk8 (*Cdk8*^2lox^*-Sox2-Cre, active in the epiblast starting from E6.5) has been generated [35] to investigate CDK8′s role in postimplantation development. Knockout embryos die at around E10.5; however, it has been shown that homozygous *Cdk8* KO female embryos displayed a greater developmental delay than males at E10.5. The reason for such gender-dependent differences in the embryonic development is that CDK8 is required for the initiation of X-chromosome inactivation by *Xist* and the recruitment of PRC2.

The normal development of the embryo depends not only on the embryo quality directly, but also on its interactions with the maternal organism. Normal placenta formation is crucial for gestation. Thus, it has been shown that *Ccnc* homozygous knockout embryos die at E10.5. Severe developmental retardation of mutant embryos and an underdeveloped placental labyrinth layer were observed [9]. Although the reasons for early embryonic lethality in this case were not fully investigated, it has been shown that in embryonic fibroblasts Cyclin C plays an important role for the entry of cells into the cell cycle and in cell proliferation.

MED1 is another component of the Mediator complex, which shows abnormalities of organogenesis at this stage. At E10.5, embryos with constitutive *Med1* KO were anemic, but had normal numbers of hematopoietic progenitor cells, and died at E11.5, presenting circulatory system impairment. It has been shown that the MED1 protein acts as a coactivator for the GATA-1 protein in the development of the erythroid cells. In the MED1-deficient mice, a significant decrease in erythroid burst-forming units and erythrocyte colony-forming units has been observed. At the same time, the myeloid lineage was unaffected [24]. A similar involvement of MED1 in hematopoiesis was noticed in adult mice with conditional KO [38].

Previously, we have mentioned that the injection of Morpholino oligos against *Med13* in zygotes resulted in the death of murine embryos at the 2–4-cell stage. However, the genetic inactivation of *Med13* also caused embryonic lethality, but at much later stages. A portion of *Med13*^−/−^ embryos developed to the blastocyst stage; however, no *Med13*^−/−^ pups were born. Autopsy indicated that embryo death took place before E12.5. The difference between these two phenotypes may be explained by the compensatory upregulation of *Med13l*, a *Med13* paralog, which develops in MED13 KOs during a rather long transcriptionally active stage of oocyte growth. However, postimplantation development is not rescued by MED13L [31].

A point mutation in the fourth exon of the *Med31* gene (l11Jus15) leads to the truncation of the translated protein by 26 amino acids and, as a result, to its degradation, functioning like a knockout [8]. Such homozygous embryos develop normally up to E8.5. Starting from E8.5, *Med31* mutants display progressing developmental delay, resulting in embryonic death at about E18.5. The molecular mechanism of this phenotype is associated with decreased level of CDK1, which is required for cell cycle progression after S-phase entry. The *Med31* mutant embryos also developed delayed chondrogenesis due to the absence of *Sox9* and *Col2a1* expression in the limbs of germs.

### 2.4. After Birth

The roles of the MED1 Mediator protein in postnatal development are the most studied among the rest of the proteins of the Mediator complex. Tissue specific conditional deletion of the *Med1* gene (excision of exons 8–10 with subsequent ORF shift) in epithelial cells leads to differentiation disorders in dental epithelial stem cells, resulting in the formation of ectopic hair near the incisors from the first week of postnatal development in mice. A molecular mechanism of this phenotype is explained by the aberrant differentiation of dental epithelial stem cells and reduced Notch signaling. Specifically, a high expression level of the stem cell marker *Sox2* was detected in MED1-deficient cells. Calcium signaling seems to play a decisive role in abnormal differentiation. Under normal conditions, differentiated dental epithelium participates in calcium transport from vessels to enamel. Moreover, it is known that calcium can promote keratinocyte differentiation. *Sox2* induction in *Med1* KOs leads to epithelium cells’ dedifferentiation and makes them sensitive to calcium, thus being responsible for ectopic hair growth. Intriguingly, the hair in this area could regenerate [28].

Additionally, MED1 protein plays an important role in meiosis during spermatogenesis in mice. It has been shown that *Med1* knockout in male germ cells results in the premature transition of primary spermatocytes to the zygotene and pachytene stages, which indicates an accelerated passage through prophase I of meiosis [27]. However, this does not affect the completion of meiosis or the fertility of transgenic males.

MED1 is essential for mammary gland development and lactation [30]. MED1 interacts with MED24, which plays an important role in magnifying the effects of activators on the general transcriptional machinery. Dose-dependent cumulative effects of MED1- and MED24-deficiency were observed in embryonic development and housekeeping gene expression [39]. During puberty, the *Med1*^+/−^ single heterozygous mutant mammary gland exhibited normal ductal elongation. The *Med24*^+/−^ single heterozygous mutant mammary gland also showed normal ductal development at the same age. However, the *Med1*^+/−^ *Med24*^+/−^ double heterozygous mutant mammary gland showed retarded ductal elongation during puberty. Surprisingly, *Med1*^+/−^ and *Med24*^+/−^ mammary glands showed normal physiological changes in morphology during pregnancy, lactation, and the regression stage [30]. MED1 participates in anchoring the Mediator to estrogen receptors (ER) bound to promoter regions, while the MED24/MED23/MED16 complex boosts the signal by interacting with RNA Pol II. Mammary gland-specific retardation in puberty (the effect is not observed in uteri) may be linked to a general attenuation of general transcription and DNA synthesis in luminal and basal cells, which is independent of ER-signaling.

MED1 is important for the wound healing process. In *Med1^epi−/−^* mice, whose *Med1* is disrupted under the control of the keratin 5 promoter, skin wound closure and the re-epithelialization rate were accelerated in 8-week-old *Med1^epi−/−^* mice compared to age-matched wild-type mice [29]. In this model, an elongation of migrating epithelial tongues and an increase in Ki67-positive cells at epidermal wounds were observed at the 8-week age. However, in 6-month-old *Med1*^epi−/−^ mice, skin wound healing was delayed and the number of Ki67-positive cells was reduced in comparison to control animals. This can be explained by the hair follicle bulge stem cell reduction in older KO mice [40], which leads to a decreased contribution of hair follicle stem cells to epidermal regeneration after wounding in older *Med1*^epi−/−^ mice.

### 2.5. Contradictions

As CDK8 and CDK19 are not functional without CCNC and have very low activity without MED12 [41], presumably their KOs have to phenocopy each other. Since MED13 is required for attachment of the CKM to the Mediator [42], its KO has to show a milder phenotype, switching off only Mediator-related activity. However, published data are not supportive of this hypothesis. *Med13* [31] and *Cdk8* KO [34] show the most severe phenotypes of Mediator-KO embryos dying at preimplantation stages. *Ccnc* KO die at E9.5, displaying significant retardation [9] similarly to *Med12* [22]. Several explanations are possible. First, in vitro cultivated embryos, especially those obtained by superovulation, may be of lower quality and can develop slower in suboptimal conditions. The general effect caused by experimental procedures can be even more pronounced in KO embryos. Secondly, three of four CKM proteins have paralogs, which must be taken into consideration in KO studies. For example, *Med13l* upregulation can partially compensate for the *Med13* KO. A third possibility implies that CKM proteins can have functions independent from the Mediator. For example, it was shown that MED12 regulates hematopoiesis individually [43]. 

## 3. Metabolism

Certain Mediator subunits play an important yet mechanistically unelaborated role in metabolism regulation in mice, as established by numerous knockout studies. Being part of the transcription machinery, they serve as a liaison, coupling nutrition signals with metabolic effects and specific transcription factors with the transcription activity.

The involvement of mediator subunits (MED1, MED13, CCNC, MED19, MED20, and MED23) in the lipid and carbohydrate metabolism is one of the most pronounced and well-studied factors in KO models (Table 2). Moreover, when the Mediator subunits were first discovered they were named hormone receptor and nutrient factor interacting proteins—TRAPs (thyroid hormone receptor associated protein) or DRIPs (vitamin D receptor interacting protein), as in MED1 (TRAP220/DRIP205), MED14 (TRAP170/DRIP150), MED24 (TRAP100/DRIP100), and some others [44,45].

Since the liver, fat, and muscle are key regulators of energy homeostasis, to elucidate the Mediator role in metabolism in this section we reviewed Mediator subunit knockout studies of these organs.

### 3.1. Liver Knockouts

Mice with liver knockouts (LKOs) of the Mediator components tend to obtain phenotypes with higher general insulin sensitivity and glucose clearance along with the facilitated activation of the hepatic energy expenditure program; hence, they are resistant to obesity and the development of fatty liver.

It is well recognized that MED1 is involved in general lipid, glucose, and energy metabolism regulation, adipogenesis, hepatic autophagy, and mitochondria function [49,59]. Although the MED1 subunit of the Mediator complex proves to be essential during embryogenesis, as its constitutive knockout is lethal at E11.5, inducible (iKO) tissue-specific knockout models may not manifest distinct phenotypes under normal conditions. For example, *Med1*-LKO mice do not show any abnormal development features or body mass change, nor any changes in behavior. However, neither do they exhibit any signs of hepatic steatosis under high-fat diet (HFD) conditions [46], as wild-type mice do. The protective knockout action is linked to an ineffective modulation of the Peroxisome proliferator-activated receptor gamma (PPARγ)-dependent genes. These mice also show lower glucose levels and higher insulin sensitivity, as well as resistance to general obesity.

The MED1 subunit has also been shown to play an essential role in PPARα-mediated pathways, as hepatocellular proliferation and peroxisome proliferative responses were abrogated in Med1-LKO mice [60]. B-oxidation enzymes were downregulated in MED1-knockout hepatocytes, as well. In particular, the interaction between the CREBBP cofactor and the L-PBE enzyme gene promoter was found to be destabilized. Furthermore, MED1 seems to promote hepatic autophagy and lipophagy, besides fatty acid oxidation [59].

Similarly, *Med23* LKO were also refractory to high-fat-diet-induced obesity, showing improved glucose and lipid metabolism due to gluconeogenesis deceleration, which is directly modulated by FOXO1, whose interaction with its response elements is disrupted by *Med23* deletion [58]. Noteworthily, hepatic *Med23* knockdown significantly improved the metabolite homeostasis in diabetic db/db model mice in the latter study. In addition to being required for metabolic signaling regulation, MED23 acts as an anti-inflammatory and antifibrogenic factor in the liver. Liver-ablated *Med23* mice were highly susceptible to CCl_4_ fibrosis induction compared to control littermates, while showing increased inflammatory infiltration. MED23 is suggested to regulate the RORα-directed expression of inflammatory chemokine genes *Ccl5* and *Cxcl10* through H3K9me2 dimethylation executed by histone methyltransferase G9a [61].

Additionally, Xue et al. identified similar effects of increased glucose tolerance under normal diet conditions in a *Cdk8* β-cell KO study. The tolerance was linked to alleviated pancreatic insulin secretion in CDK8-deficient cells [53]. Under normal conditions, oxysterol-binding protein related protein-3, when phosphorylated by cytoplasmic CDK8, inhibits insulin secretion. Thus, the lack of CDK8 drives the easier triggering of β-cells to secrete insulin. On the other hand, CDK8 would protect β-cells from apoptosis in metabolic stress conditions as a part of the Mediator complex—it represses neuropeptide expression in mature pancreata.

Med13-LKO mice did not exhibit any specific phenotype in fasted and fed states. Neither body or liver mass, nor insulin sensitivity, were affected. The role of MED13 as a ChREBP coactivator was determined in the upregulation of glucose-6-phosphatase. This proves that the Mediator complex kinase module is necessary for fructose and glycogen levels’ regulation in the liver [51].

It was shown that the kinase module disassembly and its dismounting from the core complex occurred in the healthy fasting mice after feeding, and was followed by the activation of the lipogenic genes’ transcription. This indicates that in this context the CKM acts as the signal sensitive Mediator repressor, which could be rapidly inactivated by mTORC1 signaling. Principally, in insulin resistant mice and diabetic db/db mice, a CKM dissociation and lipogenic genes’ transcription were detected even in the fasting state, indicating their importance for diabetic progression. mTORC1 was shown to transmit the nutrient signal required for CKM disassembly [55], while its inhibition or LKO restored the full complex and prevented transcription inactivation both in healthy and in diabetic mice. Nevertheless, it is still unclear whether mTORC1 activation is sufficient for Mediator-dependent lipogenic gene expression.

### 3.2. Adipose Tissue Knockouts

Adipose tissue Mediator subunit knockout (AKO) mice displayed hampered or even absent fat growth and differentiation (adipogenesis). White (WAT) and brown (BAT) adipose tissues responded differently to the Mediator components deficiency. Due to the inability of *Med* subunit knockout adipose tissue to store lipids (absorbed or de novo synthesized) and grow, they become deposited ectopically in the liver, as observed in *Med1*-adipoq and *Med19* knockouts [48,56]. Thus, the phenotypes of LKO and AKO mice can sometimes be opposed regarding liver pathology.

In the early perinatal stage, *Med1*-Myf5 (BAT and early dermomyotome derivatives knockout) mice demonstrated a moderately reduced BAT mass and surface temperature compared with control littermates. Although embryonic adipocytes and skeletal muscle formed successfully in the absence of MED1, pups died shortly around the weaning age, possibly due to defects in skeletal muscle or in the parts of the brain originating from Myf5 expressing cells [62] caused by nonspecific activation. *Ucp1*, *Dio2*, and *Cox8b* genes commonly expressed in active BAT were strikingly down-regulated in knockout mice [48].

*Med1*-Adipoq mice revealed a drastic loss of WAT at around 3–4 weeks of age (at the time of weaning), and a near complete absence of it by 6 weeks of age [48], possibly indicating an inability of WAT to carry out de novo lipogenesis. These mice were glucose intolerant and exhibited unresponsiveness to insulin injections. Ectopic fat deposition was instead present in the liver, confirming the development of hepatic steatosis. MED1 was demonstrated to be dispensable for adipose development in mouse embryos, but again found to be inarguably required for postnatal adipose expansion and the induction of de novo lipogenesis genes after pups switched diet from high-fat maternal milk to carbohydrate-based chow [63]. MED1 controls the induction of lipogenesis genes by facilitating the ChREBP-dependent recruitment of the Mediator to corresponding enhancers.

Similarly, *Med19* was found to be required for white adipogenesis in knockout WAT cells, which, albeit expressing early adipogenic TFs C/EBPβ, C/EBPδ, and KLF5 successfully, failed to induce PPARγ and C/EBPα pathways. *Med19* adipose-specific knockout (Adiponectin-Cre driven) led to a drastic loss of WAT, despite a slight increase in female and older mice weight, mimicking PPARγ defect. The mice displayed liver steatosis, hyperglycemia and glucose intolerance, accompanied by hepatic insulin insensitivity and elevated serum insulin levels [56]. Inducible *Med19* adipose knockout resulted in WAT dystrophy, mediated by its macrophage infiltration and apoptosis, as well as BAT whitening, i.e., obtaining a cell morphology with a single large lipid droplet. Ultimately, MED19 is stated to maintain adipocyte gene expression by facilitating PPARγ pathway signaling, binding PPARγ to RNA Pol II, as MED1 does.

*Med20* is required for BAT development, and promotes WAT growth. In *Med20* knockout newborn mice there was hardly any interscapular BAT, while it was clearly visible in *Med20*^fl/fl^ controls. Unfortunately, activated mice with MED20 depleted in preadipocytes (Pdgfra-Cre) died shortly after birth, with cerebral hemorrhage, likely because the Pdgfra-Cre, akin to Myf5-Cre, targeted regions of the brain. Notably, heterozygous *Med20* mice (*Med20*^wt/−^) showed significantly lower susceptibility to fat-induced obesity, as well as lower inguinal WAT and gonadal WAT percentage, while *Med20*^fl/fl^ mice (as expected) developed a distinct obese phenotype. The finding implies that MED20 may act in a dose-dependent manner. MED20 was shown to bridge C/EBPβ and RNA Pol II to promote the transcription of PPARγ [57].

CCNC adipose-deficient mice demonstrated a deficiency of progenitor differentiation, development, and lipid accumulation in brown adipocytes, but not for thermoregulation. Browning (activation of thermogenic genes) was observed in the inguinal and epidermal WAT of BAT-CCNC-deficient mice. CCNC-dependent lipogenic gene expression was conducted through the activation of the C/EBPα/GLUT4/ChREBP axis, similar to MED1 [54].

### 3.3. Muscle Tissue Knockouts

Muscular and hepatic tissues can be viewed as having common behavior: they execute insulin’s command to absorb nutrients from the blood. In this way, the increased insulin sensitivity of muscle knockout (MKO) mice may result in effects seen in liver knockout mice.

*Med1*, *Med13*, and *Med23* liver- and muscle-null mice share common traits: they are unresponsive to fat diet, and have a higher glucose tolerance and insulin sensitivity [47,50,58].

*Med1*-MKO again leads to obesity resistance under HFD [47]. These mice switch towards slower and more oxidative types of muscle fibers, with higher mitochondrial density.

Med23 is required for smooth muscle and adipose tissue differentiation, but constitutive total knockout mice were embryonically lethal around 10.5 days of gestation [64,65]; thus, the studies were performed on cell cultures obtained from knockout embryos. Although no *Med23* adipose tissue-specific knockout model has been described as of today, a smooth muscle knockout study was carried out recently. *Med23* deletion in smooth muscle results in aortic dilation and aortic lumen enlargement, while showing impaired vasoconstriction [66]. The overall data suggest that *Med23* plays an important role in maintaining the balance between smooth muscle cell growth and differentiation.

Conditional knockout mice with muscle-depleted MED13 do not develop liver steatosis on the HFD due to the altered fatty acid handling by the liver, the generally higher glucose clearance, and muscle insulin sensitivity, while showing no difference in adipose tissue properties compared to HFD control littermates [50]. MED13 is here shown to act as a repressor for *Glut4*-activating transcription factors NURR1 and MEF2. The GLUT4 increase in *Med13* muscle-ablated mice leads to a higher muscle glucose uptake. Since insulin secretion is decreased under low-glucose serum concentration, the liver presents reduced lipogenesis.

Mice with knockout of miRNA-378 display an augmented cardiac expression of its target—*Med13*. They are lean and demonstrate an increase in energy expenditure and resistance to diet-induced obesity and metabolic syndrome [67]. Agreeably, the cardiac deletion of *Med13* severely increases susceptibility to diet-induced obesity and metabolic syndrome, which can be explained by the upregulation of PPARγ and sterol regulatory element-binding proteins (SREBP) in the absence of MED13 [52].

## 4. Intestinal Knockout

Even though the gut is the main entrance, and a processing and a transfer point for consumed nutrients, the Mediator’s role in it is poorly studied in mammals in vivo. The metabolic dysregulations in knockout models that take place, if any, remain overlooked. Even so, there have been papers investigating the Mediator functions in the intestine, although mainly dedicated to its kinase subunit function in differentiation processes.

In an Apc^Min^ colon cancer model, mice, the conditional deletion of *Cdk8* in the intestinal epithelium (Villin-Cre driven) enhanced tumorigenesis and also led to overall lifespan shortening. The knockout did not affect the β-catenin level in vivo, but decreased the transcription of its downstream pathway targets in colonic epithelium culture. Histone H3K27me3 repressive labeling was pronouncedly reduced at promoters of several oncogenic genes in *Cdk8* KO cells, consistent with a promoter-binding decrease in EZH2, an H3K27 methylase. Notably, EZH2 levels remained unchanged, implying that CDK8 acts upstream of H3K27 methylation. Nevertheless, an H3K4me3, a mark of activation, was also reduced in some of these genes, coherent with an overall slight loss of their transcription, while mRNA levels of a few others, such as *Bmp2*, *Wnt3*, and *Heyl*, did show a significant increase [68].

*Cdk8/19* conditional knockout in intestinal brush border cells, driven by Villin-Cre, led to slightly impaired cell function and growth, but did not affect differentiation ability or organoid growth. The goblet cell count and mucus accumulation and secretion were significantly increased. The effect is thought to be linked to CFTR and solute carrier expression profile alteration [69].

Similarly, a comparable *Cdk8/19*-KO mouse model demonstrated that Mediator kinases are necessary for intestinal secretory cell lineage differentiation, but that these effects may be compensated to maintain intestinal stem cells and basal intestinal viability. Paneth cell, goblet and tuft cell counts were decreased, while the number of absorptive cells significantly increased. It was shown that CDK8 and CDK19 allowed secretory lineage differentiation by upregulating *Atoh1* expression with an ARID1A-defined enhancer upon interaction with SWI/SNF [70]. Thus, this model confirms that the Mediator functions are not limited to transcription modulation, but take part in chromatin remodeling.

## 5. Hematopoiesis and Immunity

Hematopoiesis is a multistage process of gradual commitment of a multipotent HPSC into terminally differentiated cells of lymphoid (NK cells, and T- and B-cells) or myeloid (thrombocytes, erythrocytes, monocytes/macrophages, granulocytes, and mast cells) lineage. 

Differentiation stages are tightly controlled by corresponding transcriptional programs as well as the functioning of terminally differentiated cells (e.g., switch between M1 and M2 macrophages, T-cells migration, activation and differentiation, etc.). Several Mediator complex proteins were found to contribute to this transcriptional regulation (Figure 3, Table 3).

MED12 is engaged in the earliest stage—HSC cell maintenance. Specific *Med12* knockout caused bone marrow aplasia and lethality in mice [43]. Surprisingly, it remained the only severe symptom in mice with ubiquitous MED12 KO. The underlying cause of HSC failure was the loss of the active chromatin state of the specific hematopoietic enhancers, mainly the c-Kit enhancer. It was shown that MED12 interaction with the histone acetylase p300 maintained the active state of these enhancers [43]. Another member of the CKM, CDK19, was reported to contribute to the proliferation and self-renewal ability of HSCs by attenuating the p53-dependent transcription of cyclin-dependent kinase inhibitor p21. However, CDK19^−/−^ mice had no significant phenotype and had normal cell counts in bone marrow and spleen, for RBC, WBC, and platelets [25].

Erythroid lineage was shown to be specifically affected by the *Med1* KO. Transcription at the β-globin locus was suppressed in the absence of MED1 due to the loss of GATA-1 recruitment [38]. An important outcome in terms of possible therapeutic outcome research unveiled the role of *Med1* in the M1/M2 switch of the macrophages [71]. M1 inflammatory macrophages have atherogenic properties, while M2 are considered antiatherogenic. The M2 transcriptional program is controlled by TF PPARγ, which needs MED1 for its action. This is in agreement with the fact that *Med1* knockout appeared to be atherogenic, and *Med1* overexpression had a protective effect [71].

The adaptive immune system requires tight control to balance between protection against pathogens and tumors, and autoimmunity. This control is partially fulfilled by transcriptional programs, and the Mediator complex plays a significant role in it. *Med1*, *Med23*, and *Ccnc* tissue specific knockouts demonstrated significant changes in T-cell behavior.

*Med1* knockout led to a reduced number of CD8+ cells at the periphery [72], and a deficiency of killer receptors (lectin-like receptor G1) and expression of inhibitory receptors (PD-1, TIM-3, TIGIT) in response to an acute bacterial infection [73], and higher susceptibility to apoptosis [72,73]. These changes at the transcriptional level were associated with the IL-7Rα/STAT5 pathway [72] and the *T-bet*-mediated proinflammatory transcriptional program [73]. It was shown that *Med1* was required for the transcription of the *T-bet* gene itself by transcription factor C/EBPβ [73].

In a similar way, *Med23* KO decreased the T-cell population on the periphery [74,75]. However, MED23-deficient T-cells appeared to be hyperstimulated, and caused autoimmune symptoms [75]. At the molecular level, this was explained by the fact that MED23 is required for the transcription of *Klf2* and several other negative regulators maintaining the quiescent state of T-cells [74,75].

At the same time, knockout at the bone marrow level of the *Ccnc*, one of the Kinase module proteins, led to almost opposite effects. It directed cell differentiation to the T-lineage and increased the susceptibility of the KO animals to T-ALL [12]. A proposed mechanism implied the inability of CDK8/19 in the absence of Cyclin C to directly phosphorylate ICN1 and promote its degradation, which led to constitutively active NOTCH signaling, and was unrelated to transcriptional regulation by the Mediator. However, these observations are also intriguing in the context of a recently published paper by Freitas et al. [2]. The authors demonstrated that the kinase module and remaining parts of the Mediator have oppositionally directed roles in CAR T-cells cytotoxicity, antitumor efficiency, etc. KO of the kinase module members promoted expansion and cytokine production at the transcriptional level, while KO of almost all the other Mediator proteins had a negative or nonsignificant effect. As a result, it was proposed that the CKM sterically prevents TFs from binding to the Mediator, and only upon its dissociation (or KO) can certain transcriptional programs be launched. This result was further affirmed by the finding that specific *Cdk8* KO in NK-cells increased its cytotoxicity by promoting the expression of specific genes [76]. 

## 6. Mediator Complex Subunits in Cardiomyocytes

From patient “phenotype to genotype” studies, it is evident that disturbances in the Mediator complex lead to disturbances in cardiac function [77], so corresponding studies in animal models have been particularly promising. Cardiomyocyte-specific mouse models with conditional cardiomyocyte-specific KO of *Med1*, *Med12*, *Med30*, and *Cncc* genes and point mutations in *Med30* are described (also summarized in Table 4). The deficiencies of various Mediator subunits in mouse cardiomyocytes have similar phenotypic manifestations: animals develop cardiomyopathy, myocardial fibrosis, and decreased heart contractility, leading to early mortality. Different authors have described similar mechanisms behind this phenotype, based on disturbances in gene expression patterns, although the specific genes that are regulated by Mediator KOs could differ [77,78,79,80].

In mouse models with a cardiomyocyte-specific *Med1* KO, transcription was significantly impacted (highly expressed genes were suppressed, while low-expressed genes were activated). The life span of these mice did not exceed 7 weeks; at the age of 3 weeks, the hearts were already significantly hypertrophied, and had features of fibrosis and reduced contractility. Interestingly, H3K27 histone acetylation was highly correlated with *Med1*-dependent differential gene expression. *Med1* deficiency led to ubiquitous H3K27 histone trimethylation and a decrease in the relative abundance of transcripts [78]. *Med12* cardiomyocyte-specific deletion caused the early onset of dilated cardiomyopathy in young mice, and a subsequent decrease in the heart rate and abnormal electrical activity of the heart, and in particular the prolongation of the QRS interval. At transcriptional level, *Med12* deletion led to changes in the gene expression profiles (suppression of *Atp2a2*, *Gja1*, *Gja5*, *Kcnn1*, *Pln*, *Ryr2*, and *Tnnt1*; activation of *Cacna1d*, *Casq1*, *Gja3*, and *Slc8a2*), the products of which are involved in calcium metabolism in the heart, which in turn led to a change in the contractility of cardiomyocytes [81]. *Med30* was required for the maintenance of a regulatory network specific for cardiomyocyte genes (*Mycn*, *Hey2*, *Tnni2*, *Mhy7*, *Gja1*, *Gaj5*, and *Pln*) [79]. In addition, RNA-seq showed that *Ccnc* cardiomyocyte-specific KO mice had a suppressed expression of genes involved in PPAR signaling pathways (*Pck1*, *Plin1*, *Plin5*, *Acadm*, *DBI*, and *Slc27a1*), FoxO signaling pathways (*Rag1*), and AMPK signaling pathways (*Adipoq* and *Scd1*). Interestingly, adult-expressed genes associated with hypertrophic and dilated cardiomyopathy (*Myl2*, *Ryr2*, *Tnni3*, *Tnnt1*, and *Tnnc1*), muscle contraction (*Itga7* and *Cacna1d*), and adrenergic signaling in cardiomyocytes (*Adra1b* and *Cacna2d1*) were also downregulated in hearts of *Ccnc* KO mice [80].

Many of the described mutations and deletions of the mediator complex disrupted in one way or another the normal mitochondrial homeostasis. For example, *Med30* deletion in adult cardiomyocytes led to a decrease in the expression level of a number of mitochondrial genes [79]. The I44F substitution in MED30 also led to a decrease in the ability of mitochondria to carry out oxidative phosphorylation. This corresponds with respiratory chain dysfunction and a decrease in mRNA levels of genes encoding oxidative phosphorylation enzymes. Interestingly, by keeping these mice on a ketogenic diet after weaning age, the pathological phenotype development could be arrested [82]. The overexpression of Cyclin C in cardiomyocytes causes the fragmentation of mitochondria, which indicates an increase in the dynamics of fission compared to fusion, which was also observed in cardiomyocytes after myocardial infarction. Meanwhile, *Ccnc* cardiomyocyte-specific deletion in mice led to an increase in the size of mitochondria, but with reduced metabolic function. These data confirm the findings of altered heart function in 12-week-old mice [80].

## 7. Bone Homeostasis

Vertebrate skeleton maintenance and repair is carried out by a closely interacting group of cells—mainly osteoblasts derived from mesenchymal stem cells (MSCs), which synthesize components of the bone matrix, and osteoclasts of monocytic origin that break down the bone tissue. The balance between osteoblastic and osteoclastic activity is maintained, among other things, by transcriptional mechanisms controlled by the Mediator.

*Cdk8* is highly expressed in aged MSCs and is functionally essential for bone remodeling through controlling bone resorption, at least in part, in the trabecular bone in young mice, through its expression in MSCs (Prx1-Cre activator). *Cdk8* KO in MSCs led to decreased osteoclastic activity and subsequent high bone mass, in contrast with *Cdk8* KO in osteoblasts (Col1a1-Cre activator) or osteoprogenitors *(*Osx-Cre activator). It is known that CDK8 phosphorylates transcription factor STAT1, which in turn controls the expression of the *Tnfsf11* gene, which encodes pro-osteoclastogenic cytokine RANKL, a key factor for osteoclast differentiation and activation. *Tnfsf11* (RANKL) was markedly decreased in *Cdk8* KO MSCs, whereas anti-osteoclastogenic *Tnfrsf11b* was significantly increased. This suggests that the CDK8-STAT1 pathway in MSCs may be critical for the external regulation of osteoclastogenesis [83].

It is known that aging MSCs have a higher ability to support osteoclastogenesis and a reduced ability to form osteoblasts [84]. CDK8 deficiency in senescent MSCs has also been shown to result in decreased osteoclast formation and STAT1 phosphorylation. These results indicate that the aging-associated increase in osteoclast formation is associated with an increase in *Cdk8* expression in the MSCs. It is also known that ovariectomy leads to osteoporosis and a decrease in cancellous bone volume. In mice deficient in CDK8 in MSCs, ovariectomy-induced bone loss and osteoclast activation were shown to be significantly suppressed. What is of more importance is that pharmacological inhibition had the same effect. These results indicate that CDK8 may affect the transcription of certain genes through its kinase activity, independently of the Mediator [83].

*Runx2* is the master transcription factor essential for osteoblast differentiation. In humans, mutations in *RUNX2* may be the cause of clavicular dysostosis, osteosarcoma, heritable skeletal disease, which is typically characterized by open or delayed closure of calvarial fontanelles, and clavicle hypoplasia [85]. This phenotype can be reproduced in mice with *Runx2* mutations. It has been shown that the homozygous deletion of *Med23* in MSCs and osteoprogenitors (Prx1-Cre and Osx-Cre-ER) leads to similar phenotype development in genetically modified mice. In particular, KO mice have skeletal defects in newborn pups, reduced body weight and length, thinness and underweight, impaired bone ossification, reduced bone formation, osteopenia, and decreased bone density. MED23 deficiency did not alter *Runx2* expression, but instead caused the suppression of multiple *Runx2*-regulated genes such as *Osx* and *Ocn*. These results indicate that MED23 is required for *Runx2*-dependent transcription. While the osteoblast development was impaired, no manifestations in the differentiation of osteoclasts were observed. Additional observation shows that the deletion of *Med23* further exacerbates the defective skeletal phenotype in *Runx2*^+/−^ mice [86].

## 8. Neurogenesis and Brain Function

Patients with point *CDK8* mutations showed a phenotype of intellectual disability and brain development abnormalities [87]. Brain abnormalities were also reported for MED17 [88], MED12 [17,89], and MED13 [90]. In humans, *MED23* missense-mutations are associated with mental challenges [91]. For example, R617Q substitution disrupted the ability of MED23 to interact with enhancer-bound transcription factors TCF4 and ELK1, resulting in the dysregulation of immediate early genes *JUN* and *FOS* necessary for brain development and neuroplasticity [92]. This indicates that components of the Mediator complex are critically important for the development and function of the brain.

*Med23* is highly expressed in active neuronal stem cells and neuroblasts. *Med23* KO in neural stem cells (NSCs) facilitated by the inducible specific Nestin-Cre/ER activator has been shown to be involved in hippocampal neurogenesis in adult mice. Hippocampal neurons in *Med23* KO mice had a shorter cell cycle length, while the number of neuroblasts, immature and mature newborn neurons decreased. Furthermore, the hippocampus *Med23* KO mice also had impaired spatial and contextual fear memory [93].

The Notch pathway is associated with the increased expression of cell proliferation genes (*Sat1*, *Tnf*, *Igf1*, *Cd37*, and *Ereg*) and is activated in the absence of MED23, thereby triggering the proliferation of neuronal stem cells (NSCs). This may explain the reduced differentiation of NSCs in *Med23* KO mice. *Med23* NSC KO mice had reduced expression of early response genes (*Egr1*, *Egr2*, and *Egr4*), which led to memory impairment [93].

Mutations in the *Med12* gene have also been described in association with intellectual disability [94]. Normally, MED12 interacts with SOX10, which is important for both developing and mature central nervous systems, glia, and the regulation of oligodendrocyte differentiation [95]. This interaction makes MED12 an important regulator of multiple nervous system functions.

*Med12* astrocyte KO mice demonstrated rapid hearing loss, stria vascularis degeneration, and a loss of cell–cell contacts between basal cells adjacent to the spiral ligament and their more diffuse layout. MED12 directly interacts with SOX10, and this interaction is required for the induction of *Egr2* expression. EGR2 is one of the transcription factors governing myelination during terminal differentiation of oligodendrocytes [95]. *Tjp1*, *Cdh1*, and *Gjb3* genes are critical for normal cochlear function, and their expression was also impaired in the absence of MED12. These changes in gene expression affected cell adhesion between the basal cells and the spiral ligament, possibly preventing the formation of normal endolymph and culminating in the loss of hair cell mechanotransduction [96]. The findings are once more abstracted in Table 5.

## 9. Conclusions

Thus, involvement of the Mediator was shown in in vivo models for several processes: firstly for embryonic development and metabolic regulation, but also for cardiac functioning, immune response, immune cell migration, and osteogenesis. Transcription factors associated with metabolism, lipogenesis, and glucose uptake (PPARα [49,60,80] and PPARγ [46,47,48,52,56,57,71], C/EBPs [47,48,54,57,73], ChREBP [51,54,63], and FOXO1 [51,58,80]) were frequently identified as Mediator partners and downstream regulators, even in situations unrelated to metabolism, e.g., HSC maintenance [43] and T-cells cytotoxicity [73]. These findings further unveil the role of the Mediator as a commutator, different members of which are responsible for the perception of different signals, and which launches specific transcription programs in response to these stimuli. Of certain interest is the fact that a number of key transcription factors, identified as Mediator-dependent in cell models, are not yet identified as affected in animal models (TGF-beta/SMAD, androgen and estrogen receptors), or affected only in embryonic development (such as the Wnt pathway [22,32]). This fact could reflect differences between transcriptional regulation in cell/organoid culture and in the whole organism, or in tumor/embryonic cells and in normal/differentiated cells.

It was shown that the CKM in several situations acts in opposition to the core Mediator [2,9]. This could be explained by the fact that some components (CDK8/19, MED12) of the Mediator have functions which are independent of the whole complex [9,10,11,12,25,43]. For example, CDK8/19 phosphorylates several targets outside of the transcription machinery, and the Mediator itself. However, there is also evidence that the binding of the CKM represses Mediator-dependent gene expression, whereas its dissociation promotes it. De Freitas et al. [2] proposed that CKM binding sterically prevents TFs from binding to the Mediator. 

Several CDK8/19 inhibitors were recently tested as potential anticancer drugs in acute myeloid leukemia, prostate cancer, and other diseases [13]. In the context of diverse Mediator roles in biological processes, it could be of interest to test CKM inhibitors as potential drugs against different types of pathologies: metabolic disorders, osteoporosis [83], autoimmunity and atherosclerosis. A major point of consideration is how fully these small-molecule inhibitors phenocopy the beneficial and adverse effects of genetic knockouts. While most reports of CDK8/19 antagonists show low levels of adverse effects, some of these effects correspond to observed KO phenotypes [97], although they are not shown for all inhibitors [98]. Studies comparing knockout models, chemical inhibitors and kinase-dead substitutions would help to deconvolute these discrepancies. Future investigations carried out in the models described in this review would also surely uncover additional processes controlled through the Mediator in the whole organism, hinted at by the in vitro studies.

## Figures and Tables

**Figure 1 ijms-24-09330-f001:**
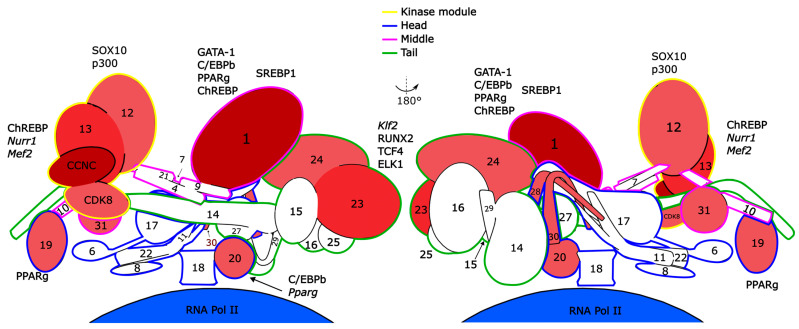
Schematic depiction of the mammalian large Mediator complex (compilation based on PDB molecular structures 6W1S, 7NVR, and 7KPX) in 180° projections and its main interactions with TFs, proteins, and genes discussed in the review. Numbers correspond to the Mediator subunit names. Proteins marked red are discussed in the current article, and the fill color saturation correlates with number of papers on those.

**Figure 2 ijms-24-09330-f002:**
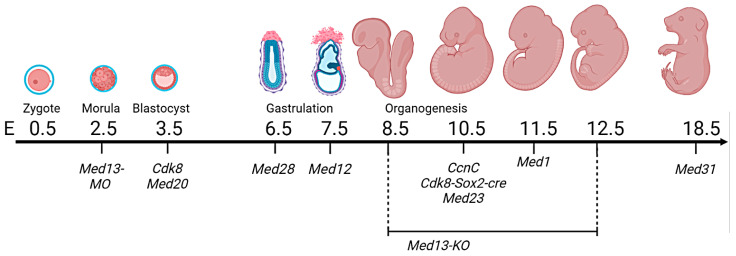
Embryonic lethality caused by Mediator subunits’ knockouts and knockdowns. “E” with a number stands for “Embryonic day #” on which the death took place.

**Figure 3 ijms-24-09330-f003:**
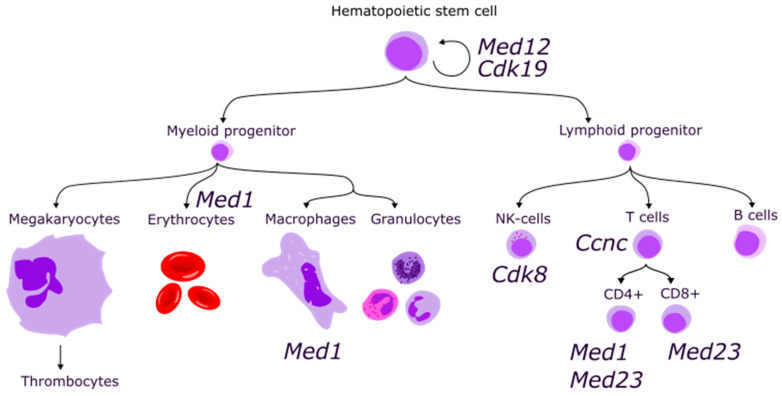
Simplified scheme of hematopoiesis and involved Mediator subunit coding genes.

**Table 1 ijms-24-09330-t001:** Effects of Mediator complex proteins’ inactivation on mouse development.

Protein	Type of Gene Inactivation	Death	Details	Reference
MED 1	Vasa-Cre embryonic germ cells		Male mice with *Med1 KO* in germ cells were fertile; however, accelerated dynamics of prophase I of meiosis was observed: germ cells prematurely appeared in zygonema and pachynema stages	Huszar, 2015 [27]
	K14-Cre ectoderm and its derivatives		Formation of “hairy” teeth due to the transformation of dental epithelium into the epidermal epithelium. Ablation of *Med1* resulted in suppression of Notch1 signaling. Suppression of Notch1 resulted in maintaining the cells in undifferentiated state, and *Sox2* expression. Calcium action on the less-differentiated cells resulted in their differentiation into the hair epidermis	Yoshizaki, 2014 [28]
	K5-Cre, epithelium		Young mice showed significantly accelerated wound closure and healing. The paracrine action of activin A was augmented due to suppressed follistatin expression, which induced phosphorylation of ERK and JNK and activation of the MAPK pathway	Noguchi, 2014 [29]
	Constitutive	E11.5 ^1^	*Med1*^−/−^ embryos at E10.5 were anemic; however, they had normal numbers of hematopoietic progenitor cells. Defects in forming erythroid burst-forming units and colony-forming units in these embryos were observed. MED1 interacted with the erythroid master regulator GATA-1. MED1 deficiency led to a defect in GATA-1-mediated transactivation	Stumpf, 2006 [24]
MED1/MED24	Constitutive, double heterozygous		Pubertal mammary glands showed significant developmental retardation. The Mediator partially failed to transduce signal from estrogen receptor bound to promoters to its target. The effect was not observed in pubertal uteri, nor in gestating glands	Hasegawa, 2012 [30]
MED12	Constitutive	E7.5	Impaired mesoderm formation. Many embryos were arrested at pre-streak stages. Wnt/β-catenin signaling was abrogated	Rocha, 2010 [22]
	Hypomorphic	E9.5	Various morphological defects Wnt/β-catenin signaling pathway was abrogated	Rocha, 2010 [22]
MED13	*Med13-MO* knockdown by morpholino injection at the zygote stage	E2.5	Delayed embryo development. Potentially, MED13 affected Zygotic genome activation, DNA reparation and mitosis progression	Miao, 2018 [31]
	Constitutive	Before E12.5	All pregnant females carrying *Med13*^Δ/Δ^ embryos lost weight between E8.5 and E12.5. No *Med13-KO* were found at E12.5	
MED 20	Constitutive	E3.5	The epiblast marker Nanog was ectopically expressed in the trophectoderm of *Med20* mutants, indicative of defects in trophoblast specification. Impaired implantation	Cui, 2018 [23]
MED23	Constitutive	E9–10.5	Mutant embryos died between embryonic E9.5 and E10.5, all three germ layers developed, and early organogenesis was initiated before death, which was likely caused by systemic circulatory failure. *Med23* KO nearly eliminated *Egr1* transcription in embryonic stem (ES) cells, leaving a paused polymerase at the promoter	Balamotis, 2009 [21]
	*Med23^sn/sn^* (*snouty* mutation, single base pair nonsynonymous mutation in Exon 22	E10.5	Craniofacial abnormalities, including malformed frontonasal prominences, pharyngeal arches, otic and optic vesicles were observed. WNT/β-catenin signaling	Dash, 2020 [32]
	*Med23^bgeo/bgeo^* (genetrap between exons 12 and 13)	E10.5	Developmental delay, failure to undergo axial turning	Dash, 2020 [32]
	Wnt1-Cre, neural crest cells	P0	Micrognathia, glossoptosis, and cleft palate were observed. *Sox9* mRNA and protein levels were both upregulated in neural crest cell-derived mesenchyme surrounding Meckel’s cartilage and in the palatal shelves	Dash, 2021 [33]
	Tek-Cre*,* vascular endothelium	E16.5	Hemorrhage and edema	Dash, 2020 [32]
MED28	Protamine-Cre	E6.5	MED28-deficiency caused the loss of pluripotency of the inner cell mass accompanied by reduced expression of key pluripotency transcription factors *Oct4* and *Nanog*	Li, 2015 [7]
MED31	Constitutive truncation	Prenatal lethality, around E18.5	Defective or delayed chondrogenesis due to a lack of *Sox9* and *Col2a1* expression. MED31 mutant embryos had fewer proliferating cells than controls. Delayed embryo development was observed after the E8.5 stage.	Risley, 2010 [8]
CDK8	Constitutive genetrap insertion in intron 4 of *Cdk8*	E3.5	Blastomere fragmentation, disrupted morula compaction	Westerling, 2007 [34]
	Sox2-Cre Epiblast	Around E10.5	Delayed embryos development. CDK8 was potentially involved in the X-chromosome inactivation	Postlmayr, 2020 [35]
CCNC	Constitutive	E10.5	Severe developmental retardation of mutant embryos, underdeveloped placental labyrinth layer	Li, 2014 [9]

^1^ “E#” stands for “Embryonic day #”.

**Table 2 ijms-24-09330-t002:** Metabolic effects of the Mediator subunits knockouts.

Gene	Activator (Name, Cell Type)	Phenotype	Mechanism (TFs)	Reference
*Med1*	Albumin-Cre, liver	Liver steatosis protection. Glucose tolerance, insulin sensitivity	PPARγ functional impairment	Bai, 2011 [46]
	MCK-Cre, muscle	Obesity resistance. Glucose tolerance, insulin sensitivity. Switch towards slow muscle fibers	C/EBPα, PPARγ; UCP-1 and Cidea↑ ^1^	Chen, 2010 [47]
	Myf5-Cre, dermomyotome derivatives at E10.5, then brown adipocytes	Growth retardation, weaning age death. Significant BAT mass reduction at E17.5	*Pparγ* impaired. *C/ebpα*, *Adipoq*. *Ucp1*, *Dio2*, and *Cox8b*↓ ^1^	Ito, 2021 [48]
	Adipoq-Cre, WAT	Nearly complete WAT loss by 6 wk age, mild decrease in BAT. Hepatic steatosis, insulin resistance		-”-
	Albumin-Cre, liver	Diminished cellular proliferation. Abrogation of peroxisome proliferative response	PPARα	Jia, 2004 [49]
*Med13*	Myo-Cre, skeletal muscle	Hepatic steatosis resistance due to improved glucose clearance (on HFD) and reduced insulin level	NURR1, MEF2, GLUT4 ↑ ^1^	Amoasii, 2016 [50]
	Albumin-Cre, liver	Unchanged insulin sensitivity. Glycogen accumulation defect when fed fructose	G6PC, ChREBP, FOXO1	Youn, 2021 [51]
	αMHC-Cre, cardiomyocyte	More susceptible to diet-induced obesity, metabolic syndrome	SREBP, RXR, PPARγ *Thrsp*, *Gpd2*, *Eno1*, *Aacs*	Grueter, 2012 [52]
*Cdk8*	Rip-Cre, pancreatic β cells	Higher glucose clearance under normal diet. Increased β cell sensitivity. Ectopic expression of neuropeptides in the cells	OSBPL3 lack of phosphorylation by CDK8 leads to insulin secretion. KO allows neuropeptide secretion to rescue cells from apoptosis.	Xue, 2019 [53]
*Ccnc*	Myf5-Cre, dermomyotome derivatives at E10.5, then brown adipocytes	BAT paucity, neonatal lethality	C/EBPα, GLUT4, ChREBP	Song, 2022 [54]
	Ucp1-Cre, BAT	Browning of WAT upon cold exposure. Less lipid accumulation in BAT on normal diet	Akt2	-”-
	Adipoq-Cre, all adipose tissue	Less lipid accumulation in BAT on normal diet		-”-
	AAV8-Tbg-Cre, liver	No phenotype described	mTORC1 pathway	Youn, 2019 [55]
*Med19*	Adiponectin-Cre, adipose tissue	Loss of WAT, whitening of brown fat, hepatic steatosis, and insulin resistance	PPARγ	Dean, 2020 [56]
*Med20*	Pdgfra-Cre, preadipocytes	Neonatal death. Severe BAT atrophy. Heterozygous mice are born normally and resist obesity on HFD. Glucose tolerance	MED20 bridges C/EBPβ to Pol II to promote transcription of PPARγ	Tang, 2021 [57]
*Med23*	Albumin-Cre, liver	Improved glucose and lipid metabolism, insulin sensitivity	FOXO1 target genes (*Irs2*, *Igfbp1*, *Pck1*, *G6pc*)	Chu, 2014 [58]

^1^ ↑ and ↓ symbols stand for increase and decrease in corresponding genes expression or protein levels, respectively.

**Table 3 ijms-24-09330-t003:** Description of hematopoiesis and immunity-linked genome-edited models of the Mediator complex.

Gene	Activator Details, Cell Type	Phenotype	Molecular Mechanism (TF)	Study
Med1	Mx-Cre, Splenocytes, bone marrow, inducible	Block of erythroid development	GATA-1, TFIIB, absence of β-globin gene expression	Stumpf, 2010 [38]
	Lyz2-Cre; myeloid lineage, macrophages; ApoE^−/−^ background	Atherosclerosis enhancement; Med1 overexpression protective	PPARγ↓ ^1^ H3K4me1 and H3K27ac↑ ^1^ at M2 marker genes	Bai, 2017 [71]
	CD8+ T cells	Expansion↓ Killer population↓ Apoptosis↑	*T-bet*– and *Zeb2* transcriptional programs↓ C/EBPβ	Jiao, 2022 [73]
	Lck-Cre T-cells	CD8+ T cell↓ in spleen	IL-7Rα↓ pSTAT5↓ BIM↑	Lei, 2021 [72]
Med23	Lck-Cre T-cells	T-cells failed to efficiently populate the peripheral lymphoid organs	*Egr1*, *Egr2*, *Cd52*↓ KLF2↓	Kasper, 2014 [74]
	Cd4Cre T-cells	T-cell hyperactivation, autoimmune syndrome	*Egr1*, *Egr2* and *Klf2*↓	Sun, 2014 [75]
Med12	Vav-Cre blood cells and HSPC Mx-Cre Splenocytes, bone marrow, inducible CreER ubiquitous	Bone marrow aplasia	p300 and CBP occupancy of c-kit enhancer	Aranda-Orgiles, 2016 [43]
Cdk19	Constitutive	Proliferation of hematopoietic stem cells under stress is impaired	P53 dependent expression of p21	Zhang, 2022 [25]
Ccnc	Mx1-Cre Splenocytes, bone marrow, inducible	T-cell-acute lymphoblastic leukemia, oncogenic Increased differentiation in T-cell lineage	ICN1 phosphorylation↓ stability↑	Li, 2014 [9]
Cdk8	Ncr1-Cre, NK-cells	Increased cytotoxicity	Expression of cytotoxic genes↑	Witalisz-Siepracka, 2018 [76]

^1^ ↑ and ↓ symbols stand for increase and decrease in corresponding genes’ expression, epigenetic marks or protein levels, respectively.

**Table 4 ijms-24-09330-t004:** Cardiac alterations of the Mediator subunit knockout and edited mice.

Gene	Activator Details, Cell Type	Phenotype	Mechanism	Reference
Med1	αMHC-Cre, cardiomyocyte	Fibrosis↑ ^1^, early lethality, cardiac contractility↓ ^1^	Deficit in RNA polymerase II recruitment to gene promoters, chromatin accessibility↓ ^1^	Hall, 2019 [78]
Med12	cKO αMHC-Cre	Dilated cardiomyopathy, cardiac contractility↓	*Atp2a2*, *Gja1*, *Gja5*, *Kcnn1*, *Pln*, *Ryr2*, and *Tnnt1* ↓; *Cacna1d*, *Casq1*, *Gja3*, and *Slc8a2* ↑—altered expression of calcium-handling genes	Baskin, 2017 [81]
Med30	cKO cTNT-Cre, icKO αMHC-Cre	Early lethality, mitochondrial cardiomyopathy	Mediator core stability (MED 4, 8, 14, 16, 17, 18, 24, 29, 31)↓ *Tnni2*, *Mhy7*, *Gja1*, *Gaj5*, *Pln, Mycn*, and *Hey2*↓	Tan, 2021 [79]
Med30	I44F protein substitution	Heart failure, fibrosis, focal myocardial necrosis, myofibril loss, progressive and selective decline in the transcription of genes necessary for OXPHOS and mitochondrial integrity, eventually leading to cardiac failure	Disturbed interactions of the ERRα, PGC-1α and Mediator complex with target genes	Krebs, 2011 [82]
Ccnc	cKO αMHC-Cre background	Heart mass↑ cardiac contractility↓	Elongated mitochondria	Ponce, 2020 [80]
Ccnc	Overexpression	Heart failure, heart hypertrophy	Mitochondrial fission ↑	Ponce, 2020 [80]

^1^ ↑ and ↓ symbols stand for increase and decrease in corresponding phenotype, gene expression or protein levels, respectively.

**Table 5 ijms-24-09330-t005:** Nervous tissue-related Mediator subunit knockout studies.

Gene	Activator (Name, Cell Type)	Phenotype	Molecular Mechanism (TF)	Reference
Med23	Nestin-Cre/ER Inducible KO in neural stem cells	Proliferation of neural stem cells ↑ ^1^, cell cycle shortening Neuroblasts and immature neurons number in dentate gyrus ↓^1^, defective dendritic morphogenesis, deficiency in spatial and contextual fear memory	Reduction in cell cycle length of NSCs with unchanged progenitors. Enrichment in genes involved in cell proliferation (*Sat1, Tnf, Igf1, Cd37*, and *Ereg*), early response genes (such as *Egr1, Egr2*, and *Egr4*) expression ↓ ^1^	Chen, 2020 [93]
Med12	Aldh1l1-Cre/ER, astrocytes	Rapid hearing loss, stria vascularis degeneration, disorganization of basal cells adjacent to the spiral ligament, stria vascularis, leading to endolymph ionic imbalance, hair cell function disorder and subsequent hearing loss	Reduced interaction between MED12 and SOX10 leads to incorrect differentiation of oligodendrocytes; *Tjp1, Cdh1*, and *Gjb3* genes expression disturbances	Huang, 2021 [96]

^1^ ↑ and ↓ symbols stand for increase and decrease in corresponding phenotype, gene expression or protein levels, respectively.

## Data Availability

No new data were created.

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
