# Peer review of "Genetically Engineered Mice Unveil In Vivo Roles of the Mediator Complex"

_ijms, 2023, doi:10.3390/ijms24119330_

Round 1

Reviewer 1 Report

Ilchuk et al. wrote a comprehensive review of the biological functions of Mediator complex components, focusing on in-vivo mouse model studies. The authors presented detailed descriptions of the mutant phenotypes of Mediator components across different developmental stages, metabolism, immunity, and neuronal functions in mouse model. In addition, they provided information on the targets of Mediator and the molecular pathways involved in each case. Given the complexity and importance of the proteins involved in gene regulation, the authors' review is well-organized and thorough. Overall, I believe that the review is valuable for readers in the field of study, and I have only minor suggestions for improvement.

1.       The introduction needs more general background about Mediators for readers who are not familiar with this topic: canonical step by step function during transcription, evolution, conservation (vertebrate vs invertebrate, eukaryote vs bacteria/archaea, etc.), and characteristics (e.g. dynamic structural shift, essential or disposable, etc.).

2.       In figure 1, label or draw a boundary of at least head, middle and tail.

3.       In the ‘Embryogenesis and development’ section, line 85, it should be CDK19-/-, not MED19-/-?

4.       Since the Med function is significant in the early stage of transcription activation, this implies that Med function is more weighed at zygotic transcription (post M-Z transition) rather than during the maternal effect stage. The authors can discuss this in the ‘pre-implantation’ section.

5.       In the ‘After birth’ section, the authors focus on Med1 deletion, largely categorized by tissue differentiation and spermatogenesis defects. This section can be improved by focusing on postnatal development, including fertility and germline development. In addition, I also suggest having a separate section for lifespan or the aging phenotype, if the authors can find that Med impacts aging.

6.       The review describes studies using predominantly the conditional knock-out mouse model to identify biological functions and molecular mechanisms, which is understandable considering the early lethality of the Med mutation. However, is there any mouse studies utilizing hypomorphic, point mutation, gain of function or even heterozygous alleles?

7.       Other than biological phenotypes, the authors should also address transcriptome and epigenetic studies using the Med mutant or CKM inhibitor, if there are any.

8.       Like other sections, I strongly suggest making a table for the ‘neurogenesis and brain function’ category as this may be a high interest topic.

Author Response

Dear Reviewer #1,

Thank you very much for your questions and commentaries. Below we will provide the answers:

  1. The introduction needs more general background about Mediators for readers who are not familiar with this topic: canonical step by step function during transcription, evolution, conservation (vertebrate vs invertebrate, eukaryote vs bacteria/archaea, etc.), and characteristics (e.g. dynamic structural shift, essential or disposable, etc.).

We significantly rewrote second paragraph of Introduction to describe the basic information about Mediator complex.

2. In figure 1, label or draw a boundary of at least head, middle and tail.

Modified.

3. In the ‘Embryogenesis and development’ section, line 85, it should be CDK19-/-, not MED19-/-?

Corrected.

4. Since the Med function is significant in the early stage of transcription activation, this implies that Med function is more weighed at zygotic transcription (post M-Z transition) rather than during the maternal effect stage. The authors can discuss this in the ‘pre-implantation’ section.

Discussed in the “Preimplantation” section, second paragraph.

5. In the ‘After birth’ section, the authors focus on Med1 deletion, largely categorized by tissue differentiation and spermatogenesis defects. This section can be improved by focusing on postnatal development, including fertility and germline development. In addition, I also suggest having a separate section for lifespan or the aging phenotype, if the authors can find that Med impacts aging.

We significantly expanded this section, however we were unable to find the information about the aging and lifespan due to the limited number of published articles in this area.

6. The review describes studies using predominantly the conditional knock-out mouse model to identify biological functions and molecular mechanisms, which is understandable considering the early lethality of the Med mutation. However, is there any mouse studies utilizing hypomorphic, point mutation, gain of function or even heterozygous alleles?

Stated point mutations have been mentioned in the text and the tables, e.g. Table 1 MED12, page 8, second paragraph MED23, page 9 first paragraph Med31, table 4 Med30. Table 1 MED1/24 information has been added, Table 2 MED20, Page 14, last paragraph in the section “muscle tissue” MED13.

7. Other than biological phenotypes, the authors should also address transcriptome and epigenetic studies using the Med mutant or CKM inhibitor, if there are any.

The current article is focused on the in vivo effects of genetically modified mice. We suppose, that discussion of CKM inhibitors is slightly out of scope and will complicate the manuscript for readers. There is no many scientific data about the action of Mediator mutants on epigenetic and transcriptome, however we have mentioned some effects in the sections Liver knockouts 4th paragraph, intestinal knockout 2nd paragraph, Mediator complex subunits in cardiomyocytes penultimate paragraph.

8. Like other sections, I strongly suggest making a table for the ‘neurogenesis and brain function’ category as this may be a high interest topic.

Table 5 has been added.

Reviewer 2 Report

The authors have submitted a nice and comprehensive review focused on the role of Mediator components in physiological processes, disease, and embryo development. Most of these roles has been obtained from genetically engineered mouse models that are well described in this review. The review updated the preceding data in this field showing existing mouse models for studying the Mediator complex. In my opinion the redaction style is attractive and understandable. 

Author Response

Thank you very much for the review.